# Mitochondrial Cardiomyopathy: Molecular Epidemiology, Diagnosis, Models, and Therapeutic Management

**DOI:** 10.3390/cells11213511

**Published:** 2022-11-06

**Authors:** Jinjuan Yang, Shaoxiang Chen, Fuyu Duan, Xiuxiu Wang, Xiaoxian Zhang, Boonxuan Lian, Meng Kou, Zhixin Chiang, Ziyue Li, Qizhou Lian

**Affiliations:** 1Cord Blood Bank Centre, Guangzhou Women and Children’s Medical Centre, Guangzhou Medical University, Guangzhou 510180, China; 2Department of Laboratory Medicine, Pingyang People’s Hospital Affiliated to Wenzhou Medical University, Wenzhou 325499, China; 3Adelaide Medical School, University of Adelaide, 30 Frome Rd., Adelaide, SA 5000, Australia; 4Department of Allied Health Science Faculty of Science, Tunku Abdul Rahman University, Ipoh 31900, Malaysia; 5Department of Surgery, Shenzhen Hong Kong University Hospital, Shenzhen 518053, China; 6State Key Laboratory of Pharmaceutical Biotechnology, University of Hong Kong, Hong Kong 999077, China

**Keywords:** mitochondrial cardiomyopathy, molecular epidemiology, diagnosis, animal model, cellular model, gene therapy, pharmacological approach, mitochondrial transfer/transplantation

## Abstract

Mitochondrial cardiomyopathy (MCM) is characterized by abnormal heart-muscle structure and function, caused by mutations in the nuclear genome or mitochondrial DNA. The heterogeneity of gene mutations and various clinical presentations in patients with cardiomyopathy make its diagnosis, molecular mechanism, and therapeutics great challenges. This review describes the molecular epidemiology of MCM and its clinical features, reviews the promising diagnostic tests applied for mitochondrial diseases and cardiomyopathies, and details the animal and cellular models used for modeling cardiomyopathy and to investigate disease pathogenesis in a controlled in vitro environment. It also discusses the emerging therapeutics tested in pre-clinical and clinical studies of cardiac regeneration.

## 1. Introduction

Mitochondrial diseases are clinically and genetically a heterogeneous group of rare disorders that invariably affect mitochondrial respiratory chain (MRC) function, oxidative phosphorylation (OXPHOS), and cellular energy production. They may present at any age and collectively affect ~1/5000 births [1]. Mitochondrial dysfunction can manifest in a tissue-specific or a multisystemic manner and often affects organs with the highest energy demands, such as the brain, skeletal muscle, eyes, and heart [1]. The myocardium is highly dependent on oxidative metabolism. Normal cardiac contractile and relaxation functions are critically dependent on a continuous energy supply. Accordingly, disturbances and impaired mitochondrial bioenergetics, with subsequent disruption of ATP production underpin a wide variety of cardiac diseases, including diabetic cardiomyopathy, dilated cardiomyopathy (DCM), hypertrophic cardiomyopathy (HCM), anthracycline cardiomyopathy, peripartum cardiomyopathy (PPCM), and MCM [2]. Advances in genomics have made it clear that there is variety in phenotypic expression. The most frequent cardiac disorders related to mitochondrial dysfunction are MCM [3]. Thus, MCM is described as a myocardial disorder characterized by abnormal heart-muscle structure, function, or both, secondary to genetic mutations encoded by the mitochondrial DNA (mtDNA) or mitochondria-related nuclear DNA (nDNA), in the absence of concomitant coronary artery disease, hypertension, valvular disease, and congenital heart disease.

Compared with other forms of cardiomyopathies, the pathogenic feature of mitochondrial genetic dysfunction in MCM is secondary to the genetic mutations encoded by the mtDNA or mitochondria-related nDNA genes. Mitochondrial dysfunction is commonly involved in a broad spectrum of heart diseases and implicated in the development of heart failure via insufficient energy metabolism, abnormal ROS homeostasis, dysfunctional mitochondrial dynamics, abnormal calcium homeostasis, and mitochondrial iron overload. The pathologic characteristics of MCM present multiple organ impairment, and the heart is one of the organs commonly affected, while other forms of cardiomyopathies do not present multiple organ impairment and do not show mitochondrial functional impairment.

In this paper, we review the complex molecular epidemiology of MCM, discuss the current methods of diagnosis, and highlight the cellular models and animal models that mimic MCM in vivo and in vitro, which will aid in elucidating a tentative pathogenesis of MCM and in developing novel therapies. We also update novel and emerging therapeutics, including pharmacological strategies, gene therapies, and mitochondrial replacement therapy and artificial mitochondrial transfer/transplantation (AMT/T).

## 2. Molecular Epidemiology and Multiorgan Clinical Expression of MCM

Over the last quarter-century, there has been tremendous progress in the genetics research that has defined molecular causes for MCM [4]. More than one thousand mutations have been identified, in various genes with varying ontologies. Genetic variants associated with mitochondrial dysfunction including mtDNA deletion, and nuclear and mtDNA variation, result in damage to the assembly of the MRC complexes, the maintenance of tRNAs, rRNAs, and mtDNA, and the synthesis of coenzyme Q10 (CoQ10), indicating the diverse molecules and pathways that cause hypertrophic, dilated, restrictive, and arrhythmogenic cardiomyopathies (Table 1). Whabi et al. [5] showed that the myocardial involvement in mtDNA diseases is progressive and, in association with other cardiovascular manifestations and risk factors (i.e., intraventricular conduction blocking, diabetes, premature ventricular contractions, and left ventricular hypertrophy (LVH)), an independent predictor of morbidity and early mortality.

Based on the prevalence of mitochondrial disease and frequency of cardiac manifestations, about 1/10,000–15,000 of the general population are affected [6]. In the pediatric population with mitochondrial disease, the incidence of MCM may be as high as 21%. The overall survival rate of patients with MCM is significantly lower than in those without MCM, with ten-year Kaplan–Meier estimates of overall survival of 18 and 67%, respectively [7]. LVH, neonatal onset, and chromosomal aberrations are independent predictors of all-cause mortality.

The main manifestations of MCM are shown in Figure 1 and include diffuse ventricular hypertrophy, with progressive diastolic dysfunction and heart failure with preserved ejection fraction, which is one of the most common clinical manifestations in mitochondrial diseases. Recent studies have also reported histiocytoid cardiomyopathy and restrictive cardiomyopathy, as well as Takotsubo syndrome [1,8]. MCM is often accompanied by multisystem manifestations with neuromuscular, endocrine, and neuro-sensorial features. Patients with neuromuscular signs show creatine kinase enzyme at a normal or slightly elevated levels, but higher liver enzyme levels are seen in up to 10% of patients. Renal features may include nephrotic syndrome, tubulopathy, tubulointerstitial nephritis, and nonspecific renal failure. Endocrinopathies include hypothyroidism, hypoparathyroidism, diabetes mellitus, adrenocorticotropic hormone deficiency, and hypogonadism. Gastrointestinal symptoms (diarrhea, constipation, abdominal pain, nausea, and chronic intestinal pseudo-obstruction) may also be present. The main ophthalmologic manifestation is retinitis pigmentosa. Sensorineural hearing loss occurs in 7 to 26% of patients, with the prevalence increasing with age.

## 3. Diagnosis

Early diagnosis of MCM is thought to be intricate and challenging because of its broad clinical and genetic heterogeneity. Although relevant diagnostic schemes have been proposed to improve detection, there remains no definitive diagnostic standard. Physicians need to maintain a high level of suspicion of MCM in patients with features and symptoms that could lead to a diagnosis of mitochondrial disorder or multisystem involvement and without a clear cause. An integrated diagnosis of MCM requires detailed genetic counselling and detection, histopathological studies, biochemical screening, cardiac investigations, including cardiac magnetic resonance (CMR), and functional assays. A flow chart for diagnosis of mitochondrial disease is shown in Figure 2.

### 3.1. Genetic Counselling and Detection

Genetic counselling is a fundamental part of the diagnostic workup and should be performed by specialized personnel. Clinicians should pay attention to the detailed history of patients with a maternal family history of disease or multisystem symptoms, and their first-degree relatives should be referred to a specific cardiogenetic clinic.

Patients should undergo at least one of the following genetic analyses:Patients with clinical presentations consistent with mitochondrial myopathy, encephalopathy, lactic acidosis, or stroke-like episodes (MELAS) should undergo screening for common point mutations on mitochondrial DNA.Whole mitochondrial DNA screening [9].Investigations for mutations using a targeted gene panel of 241 genes known to cause mitochondrial diseases, as well as the whole mitochondrial genome.Whole-exome sequencing using next-generation sequencing for nuclear DNA mutations. Detailed procedures have been described previously.High-density oligonucleotide array for large chromosomal deletions, as previously described [10].

### 3.2. Cardiac Imaging Diagnosis

Hypertrophic remodeling is the dominant pattern of MCM. The early stages of MCM are characterized by progressive diastolic dysfunction and heart failure with preserved ejection fraction. The association of LVH (with or without apical trabeculation) with systolic dysfunction has been reported as a typical evolution of MCM. Previous reports of CMR in MELAS also demonstrated pericardial effusion, increased signal with T2-weighted imaging, and late gadolinium enhancement (LGE) [11]. CMR can be used to assess the etiology of cardiomyopathy in a patient without known mitochondrial disease and may also be applied to screen for myocardial involvement in patients with known mitochondrial myopathy.

Intracellular vacuolar changes indicate an increased water content in the myocardium, resulting in a diffuse increase in the T2 signal or values in the LV myocardium. T1, T2 mapping, and extracellular volume fraction analysis have been reported to provide additional information on myocardial tissue status in patients without LGE [12]. Eletrophysiologic abnormalities are common but nonspecific in patients with MCM.

### 3.3. Muscle Biopsy and Histopathological Examination

Mitochondrial disease presents with a wide spectrum of clinical manifestations. Skeletal muscle is frequently affected and represents distinctive histological and histochemical hallmarks of mitochondrial pathology in primary mtDNA-related disease. Fresh skeletal muscle biopsy is the current gold standard for diagnosing mitochondrial diseases. Compared with skeletal muscle biopsy, cardiac muscle biopsy is more invasive and can be performed in patients with rapid disease progression or when biochemical testing in fibroblasts is performed [8]. A pathological diagnosis should be made following histochemical staining combined with microscopic observation of the mitochondrial structure and morphology after muscle biopsy. A key histological feature of MCM is ragged-red fibers (RRF), visualized using modified Gomori trichrome stains, and accumulation of abnormal mitochondria in peripheral and intermuscular, which has specific diagnostic value [8]. Nonetheless, RRF is not usually present in child-onset mitochondrial diseases and is only common in advanced cases of adult-onset mitochondrial diseases. Furthermore, partially mtDNA-encoded cytochrome C oxidase complex IV (COX) and fully nuclear-encoded succinate dehydrogenase complex II (SDH), a typical histopathological change in a muscle biopsy with mitochondrial disease, are only reported in adults after 30 years of age [13].

Electron microscopic findings are an important aid in diagnosing MCM. A distinct feature is an increased number of swollen mitochondria of varying sizes and shapes [14]. Intracellular vacuolar changes indicate an increased water content in the myocardium, resulting in a diffuse increase in the T2 signal or values in the LV myocardium. In addition, internal nuclear fiber atrophy, a large number of lipid droplets and fiber accumulation, atrophy of type I or type I fibers, fibrogenesis, glycogen accumulation, and inflammatory changes are suggestive of MCM [14,15,16].

Although muscle biopsy is currently the gold standard for diagnosis of mitochondrial diseases, its sensitivity and specificity are not 100% [17]. To make a pathological diagnosis of MCM, quantitative analysis of mitochondria within the cardiac myocytes using electron microscopy and immunohistopathological analysis with respiratory chain enzyme antibodies would be useful.

### 3.4. Systematic Physical Examination

Due to the multi-organ involvement in the majority of mitochondrial diseases, an initial careful clinical examination of several organs (including neuroimaging, hearing assessment, ophthalmologic examination, liver function tests, and serum creatinine phosphokinase), with different criteria for adults and children, should initially be undertaken, to determine whether the cardiomyopathy is an isolated pathology or part of a multisystem disease [4,18]. Cardiologists should suspect a mitochondrial dysfunction in patients who, in addition to cardiomyopathy, present with hearing loss, renal dysfunction, diabetes mellitus, and peripheral myopathy. A maternal inheritance pattern or family history of mitochondrial disease may support a suspicion of MCM [19].

### 3.5. Biochemical Screening

Biochemical screening tests for mitochondrial diseases include determination of lactate dehydrogenase (LDH), plasma lactic acid, creatine kinase, blood glucose, urine organic acids, hemoglobin A1c and creatinine and plasma amino acids, abnormal thyroid function tests, and various degrees of liver dysfunction [19,20]. Although lactic acid is a common biochemical feature of many mitochondrial diseases, it is neither specific nor sensitive.

Rigorous diagnosis of MCM requires a multidisciplinary approach. Given the absence of pathognomonic findings on cardiac imaging, a high degree of clinical suspicion is warranted. Transthoracic echocardiograms and cardiac magnetic resonance imaging can rule out coronary artery disease. Laboratory tests including LDH, lactate, and creatine-kinase should be conducted.

## 4. Disease Modeling for MCM

The heterogeneity and rarity of MCM preclude randomized clinical drug trials with standardized end-points; as a result, disease modelling using animals or cells is an essential component in the study of these diseases. MCM requires the use of an appropriate model system in order to characterize the mechanisms of disease and develop useful preclinical novel therapeutics. We reviewed the models used for studying cardiac remodeling and impaired cardiac function, including genetic and drug-induced models in vivo (Table 1) and in vitro (Table 2), and discuss the genetic and physiological suitability of each model for human MCM.

### 4.1. Animal Models

Owing to the difficulties in generating animal models of MCM, cardiomyopathy was first described in 1997 in a mouse model, by knocking out the heart/muscle isoform of Ant1 [21]. The Ant1-deficient mouse exhibited cardiac hypertrophy associated with mitochondrial proliferation, similarly to that seen in humans with MCM, demonstrating for the first time that mitochondrial bioenergetic insufficiency could lead to cardiomyopathy [22]. Richman et al. reported that Med30zg mutation mice develop progressive cardiomyopathy that is invariably fatal by 7 weeks of age [23]. Mechanistically, the Med30zg mutation causes a progressive and selective decline in the transcription of genes necessary for OXPHOS and mitochondrial integrity, eventually leading to progressive cardiomyopathy and cardiac failure [24]. Mitochondrial complex I (CI) deficiency is the most common mitochondrial enzyme defect in humans. Researchers generated a CI-deficient mouse mode with complete knockout of the Ndufs6 subunit in the heart, resulting in a marked CI deficiency that has been used for studies of pathogenesis, modifier genes, and testing of therapeutic approaches [25]. The Zhang’s study employed mice and cardiomyocytes, with inactivation of mitochondrial transcription factor A (TFAM), as a model system. They demonstrated that the inactivation of TFAM simulated the central problem of mtDNA depletion, impaired OXPHOS activity, and excessive reactive oxygen species (ROS) generation and inhibition of cell cycle activity. This ultimately resulted in lethal DCM in neonatal mice, with many of the same clinical, biochemical, and ultrastructural features found in human MCM [26,27]. Sayles et al. reported that mutant CHCHD10 and/or CHCHD2 proteotoxicity activate the mitochondrial integrated stress response (ISR^mt^), which causes profound metabolic imbalances, culminating in oxidative stress and iron dysregulation, and ultimately resulting in mitochondrial dysfunction and contributing to disease pathogenesis [28,29]. Recently, a myocardium-specific Isca1 knockout rat model of multiple mitochondrial dysfunction syndromes (MMDS) complicated with cardiomyopathy was applied to study the mechanism of energy metabolism in cardiovascular diseases, as well as for the development of drugs [30]. Nevertheless, the specific mechanisms that underlie cardiac developmental disorders are not yet well understood, and there is an urgent need for suitable in vivo animal models to aid research. In addition, dog [31] and zebrafish [32] have been used for modeling mitochondrial disease. A Mongolian gerbil model of iron overload in mitochondrial contributed to cellular oxidative stress, mitochondrial damage, and cardiac arrhythmias, as well as the development of cardiomyopathy [33], and showed similar pathophysiological changes to patients with iron overload who develop cardiomyopathy.

Animal models have also been used to evaluate potential therapies for MCM. By knocking out Frataxin in mice, researchers generated a model that recapitulated most features of Friedreich’s ataxia (FRDA)-associated cardiomyopathy. Intravenous administration of adeno-associated virus (AAV) rh10 vector expressing human frataxin prevented the development of cardiomyopathy, but more importantly, administration after onset of heart failure was able to reverse the cardiomyopathy at functional, cellular, and molecular levels within a few days [34]. CIII-deficient Bcs1l^p.S78G^ knock-in mice displaying multiple visceral manifestations and premature death, with transgenic expression of alternative oxidase (AOX), showed normal-sized hearts with no fibrosis throughout their life; evidence that AOX is able to attenuate CIII- or CIV-related pathological development, and is potentially translatable to patients with CIII- or CIV-blockade and associated MCM, using gene therapy approaches [35].

Recently, cardiac/cardiomyocyte-specific knockout models were generated, to gain an insight into the underlying etiology of cardiomyopathy. For example, TAZ cardiomyocyte-specific conditional knockout (cKO) mice were generated, and multiple physiological and biochemical aspects of BTHS cardiomyopathy were mirrored, which gave important insights into the underlying etiology of BTHS cardiomyopathy [36]. Cardiomyocyte-specific complement component 1 Q subcomponent-binding protein (C1QBP) cKO mice were used to examine the physiological function of mitochondrial p32/C1QBP in the heart [37]. Cardiomyocyte-specific-deletion-of-Crif1 mice were generated for early and late onset MCM, respectively [38], and provided insights for understanding the progression of MCM.

Animal models have contributed greatly to our understanding of mitochondrial disease. Nonetheless, with important genetic, biological, and physiological differences, animal models, especially mice models, may not reproduce human pathophysiology or recapitulate the considerable genetic variation that exists in disease populations. This has prevented the translation of such findings to human therapeutics [39]. Moreover, mice models are not suitable for large-scale toxicity screening or therapeutic molecule testing [40]. Improved models are therefore desperately needed, to understand patient-specific disease mechanisms and clinical pharmacotherapy.

### 4.2. Cellular Models

Cellular models represent an important means by which to study rare human diseases, including genetic cardiomyopathies. The discovery of iPSCs may circumvent these limitations and provide new insights into the pathophysiology of cardiomyopathies. In this paper, the different cellular models used to study MCM are illustrated in Table 3.

#### 4.2.1. Immortalized Cells

Generic immortalized cells have been used to study two inherited diseases caused by point mutations in mtDNA: MELAS syndrome [41] and MERRF (myoclonic epilepsy and ragged-red fibers) [42]. In both diseases, an alteration in the post-transcriptional modification of a uridine located in an essential position for specific mitochondrial tRNAs results in OXPHOS impairment and a subsequent inability to generate sufficient ATP to meet the energy demands of the cell. To study Barth syndrome (BTHS), the authors modeled BTHS and cardiolipin deficiency by knocking-out tafazzin (TAZ) gene in a myoblast cell line (C2C12) [43]. This BTHS myoblast model helps to elucidate the mechanisms by which defective cardiolipin remodeling interferes with normal myocyte differentiation and skeletal muscle ontogenesis. Although the establishment of immortalized cell lines has helped to study biological and molecular events, these might not be the best tool for accurately recapitulating the disease.

#### 4.2.2. Fibroblasts

Fibroblasts have also been used to study MELAS, MERRF [44], and BTHS [45], to study the pathology of mitochondrial disease, and to help understand its molecular basis. The fibroblasts from pediatric patients were used to correlate disease severity with cellular lipid abnormalities and to gain an insight into the phenotypic complexity of the disease. Although these studies successfully used fibroblasts to analyze different MCMs, all studies had to work within the limitation of fibroblast passage number. Decreases in the number of mitochondria or changes to the structure of organelles should nonetheless also be taken into consideration when drawing conclusions.

#### 4.2.3. Induced Pluripotent Stem Cells (iPSCs) and iPSC-Derived Cardiomyocytes (iPSC-CMs)

Over the last decades, the rapid development of human iPSC technology has facilitated the study of human diseases using patient-specific cells in vitro. The capacity of iPSCs to retain patient-specific information, including genetic and epigenetic characteristics, makes them suitable for modelling cardiac diseases [56]. iPSCs have excellent potential to revolutionize the study of rare disease mechanisms and to screen for efficacious drugs in human tissues [57]. Cardiac disease was the first case in which iPSCs from patients were used [46]. Given that the genetic background for an individual is preserved, the use of these patient-specific cells represents perhaps the best tool to realize personalized medicine [58]. Studies reported that iPSC-derived fibroblasts recapitulated the disease phenotype and did not demonstrate altered levels of heteroplasmy in culture and, therefore, represented a unique and novel model [47,48]; however, appropriate heteroplasmy levels, mitochondrial maturation, and mitochondrial respiratory functions should be considered. Moreover, significant progress has been made in developing efficient protocols for the directed differentiation of iPSCs into functional cardiomyocytes (iPSC-CMs); thus, we are now able to generate defined cardiac subtypes in large quantities and with high purity [59]. There is substantial evidence showing that iPSC-CMs with congenital cardiomyopathies represent the phenotype of the disease, such as abnormalities in cardiolipin processing, sarcomere assembly, myocardial contraction, ROS production, and cardiomyocytes functioning, and correlating with the abnormalities and cardiac dysfunction observed in patients [49,50]. This shows, once again, that iPSC-CMs provide insight into human disease and can be used to test potential therapies, thereby providing a unique and powerful platform for modeling genetic diseases, as well as for investigating the underlying pathological mechanisms.

Patient-derived iPSC-CMs are now used in the study of human BTHS [49], FRDA-associated HCM [51,52,53,54], and dilated cardiomyopathy with ataxia syndrome (DCMA) [55], given their shared genomic and transcriptomic profiles. In a unique in vitro model system, mitochondrial fragmentation was significantly rescued by the mitochondrially- targeted peptide SS-31 [55], suggesting another novel therapy for MCM.

## 5. Advantages and Limitations of iPSC/iPSC-CM

Research using iPSCs has several advantages, one of which is the availability of required starting materials. iPSCs can be generated from many different somatic cell types, including fibroblasts and peripheral blood mononuclear cells (PBMCs), that are easily accessible and require minimally invasive procedures. iPSCs contain the genetic makeup of the individual from whom they were generated, and the underlying predisposition towards any disease can be recapitulated in a petri-dish. More importantly, human iPSC-CMs offer a far more accurate platform for phenotypic/target screens and cardiotoxicity studies. In addition, experimental studies with iPSC-CMs can be undertaken in parallel with clinical trials, to gather additional information, and may have a direct influence on the related clinical study. These features make iPSCs an ideal platform to study the pathophysiology of cardiac diseases. Nonetheless, there are some limitations. First, as observed previously, the culture environment, including the undefined nature of serum and associated batch batch-to-batch variations, may have profound effects on disease phenotype and on the compounds in drug-related studies [60]. Second, it is important to note that the genetic/epigenetic makeup of individual iPSCs may be influenced during the reprogramming process and culture conditions. This may lead to unexpected phenotypes in iPSC-derived progenies [61]. Third, although recent studies have reported iPSC-CMs to be a suitable experimental model for mitochondrial disease, they are limited by their poor mitochondrial bioenergetic capacity when compared with adult cardiac tissue. This may have relevance in disease modeling, particularly of mitochondrial disorders. Finally, the physiologically immature state of iPSC-CMs might be a critical limitation in cardiac disease modeling, by masking disease-relevant phenotypes, such as the disease-associated bioenergetics of mitochondrial function. The circulating factors and drugs tested as potential inductors of iPSC-CM maturation should also be taken into consideration in iPSC-based disease modeling of MCM.

## 6. Current and Novel Management in MCM

Due to the extreme heterogeneity of MCM and a multiorgan phenotype, its management faces very particular challenges, including the general treatment of cardiomyopathy and the symptomatic treatment of complications. Progress has been made in the current and emerging therapeutic landscape, including novel pharmacological treatments, gene therapy, mitochondrial replacement therapy, and AMT/T in the treatment of MCM. We summarize the pharmacological and non-pharmacological therapies, from a bench to bedside perspective, that may become mainstream treatments in the coming years.

### 6.1. Pharmacological Strateges

Current therapies for MCM comprise standard heart failure therapy and several pharmacological agents aimed at facilitating mitochondrial biogenesis and maintaining mitochondrial function, to prevent cardiac dysfunction, including (i) enhancing mitochondrial biogenesis; (ii) retaining mitochondrial metabolism by limiting fatty acid import and oxidation, while improving glucose utilization; (iii) reducing ROS production using ROS scavengers; and (iv) inhibiting mitochondrial permeability transition pore (mPTP) with the use of mPTP inhibitors [62].

Polyphenol compounds such as resveratrol and epicatechin activate SIRT1/3, to promote adenosine monophosphate-activated protein kinase (AMPK) and endothelial nitric oxide synthase (eNOS), which are responsible for regulating mitochondrial biogenesis through the activation of mitochondrial transcription factors, PGC1α, and nuclear respiratory factors (NRFs) [63]. Nicotinamide riboside (NR) can be utilized as a diet supplement, to enhance nicotinamide adenine dinucleotide (NAD^+^) availability, leading to SIRT1 activation. NR-supplementation has been shown to improve NAD^+^ levels, thus attenuating the progression to heart failure in mouse models of both DCM and cardiac hypertrophy [64]. Published studies have shown that stomatin-like protein-2 (SLP-2, a mitochondrial-associated protein abundant in cardiomyocytes) protects the mitochondria by stabilizing the function of optic atrophy 1 (OPA1), promoting mitofusin (Mfn) 2 expression, interacting with prohibitins and cardiolipin, and stabilizing MRC complexes, suggesting that SLP-2 is a potential target for the treatment of MCM [65]. Nonetheless the specific mechanism of SLP-2 needs to be confirmed in further research. Reducing ROS production is a theoretical strategy to treat cardiac pathological remodeling and heart failure. MitoQ, a ROS scavenger linked to triphenylphosphonium, is the best-characterized mitochondria- targeting antioxidant. By accumulating in the negatively charged mitochondrial matrix, lipophilic triphenylphosphonium cations of MitoQ effectively recruit CoQ10 to the mitochondrial matrix, in order to improve the ETC efficiency and reduce ROS production. In preclinical studies, MitoQ appeared to be protective in animal models with hypertension, I/R injury, and pressure overload stress [66]. In a small clinical trial involving 55 middle-aged and older adults, MitoQ treatment reduced aortic stiffness, plasma oxidized LDL, and improved vascular endothelial function [67]. In addition to MitoQ, other ROS-scavenging compounds, such as EUK-8 [68], XJB-5-131 [69], SS-31 [55,70], and Mito-tempo [71] were shown to protect mouse hearts against pressure overload-induced heart failure and DCM. However, since the studies on the above compounds are mostly based on transgenic animal models, further clinical investigation into their safety and efficacy is required. Inhibition of mPTP presents a promising therapeutic strategy to prevent cardiomyocyte death and mitochondrial dysfunction. CsA, a CypD inhibitor, has previously been shown to protect the Langendorff-perfused hearts from re-oxygenation and perfusion injury [72], and to enhance mitochondrial respiratory function with heart failure [73].

Mitochondrial dysfunction contributes to the development of cardiomyopathies through energy deprivation, the accumulation of ROS, and cell death. Pharmacological strategies aimed at maintaining mitochondrial function are challenging, due to the different etiologies of MCM.

### 6.2. Gene Therapy for MCM

Gene therapy is a novel approach to restoring the expression of critical proteins responsible for mitochondrial function via non-viral approaches or viral approaches, with the latter including lentiviral vectors, adenoviral vectors, and adeno-associated virus serotype (AAVs) delivery systems. AAVs are preferably used over other viral vectors because of the low risk of random insertion into the nuclear genome and their long-term persistence in cells. AAVs have been used to deliver therapeutic genes in several mouse models of MCM. rhAVV 10-based gene therapy has been successfully achieved in a mouse model of FRDA, and the results showed that the administration of the AAV-frataxin vector was able to completely reverse the cardiomyopathy of mice [34]. AAV9 can efficiently deliver a transgene under the control of a cardiac-specific promoter into cardiomyocytes. Liu et al. reported that ERK5 is a requisite for sustaining PGC1α expression, and the restoration of ERK5 expression by the AAV9 system ameliorates mitochondrial function and prevents high-fat-diet-induced cardiomyopathy [74].

AAV-based gene therapy represents the most straightforward precision-medicine approach for many mitochondrial diseases, although several issues must be considered for its successful use. Most MCM are multi-systemic syndromes; therefore, a widespread gene expression of the vector carrying the therapeutic transgene is required to achieve a significant recovery of the patient’s condition. However, delivering and expressing an ectopic gene throughout the whole body remains challenging. Genetic therapies are still far from becoming routine for MCM, due to the technical and regulatory issues that make this intervention extremely expensive. Given the promising advances in viral vector and gene editing technologies, we expect considerable benefits will be achieved for varied individual MCM, using the best routes of administration and precision medicine.

### 6.3. Mitochondrial Replacement Therapy and AMT/T

Mitochondrial replacement therapy is a method in which nuclear DNA from a mother with a mtDNA mutation is transferred to an oocyte or zygote that contains normal mtDNA from a healthy donor [75,76]. There have been several reports showing the feasibility of these techniques in human oocytes and that good quality embryos can be produced [77]. These studies are crucial for determining the possible advantages of mitochondrial transplantation in mitigating mitochondrial diseases. However, due to the lack of regulation of in vitro fertilization (IVF) techniques in many countries, mitochondrial replacement therapy is being used in unregulated environments and in some cases without appropriate ethical approval.

Multiple research groups have developed AMT/T methods that transfer healthy mitochondria into damaged cells to recover cellular function [78,79,80]. A considerable amount of data has been accumulated on the therapeutic effects of mitochondrial transplantation in the heart [81,82], liver [83], brain [84], eye [85,86], kidney [87], lung [78,88,89], nerve [90,91], and skeletal muscle [92]. It has been reported that transplantation of mitochondria into cardiomyocytes leads to a short-term improvement of bioenergetics [93,94] and viability [95], enhancement of antioxidant capacity [96], and a reduction of apoptosis [97], indicating a supercharged state, while the improved effects disappeared over time. Efficacious mitochondrial transfer from iPSC-MSCs with high intrinsic Rho GTPase 1 (MIRO1) and sensitivity rescued anthracycline-induced cardiomyopathy [97]. Human mitochondrial transplantation has been performed in a clinical setting. Emani et al. transplanted mitochondria isolated from patients’ muscles into ischemia-reperfusion-injured hearts. The results showed that ventricular function improved in all patients, and four out of five subjects successfully separated from extracorporeal membrane oxygenation (ECMO) support [98]. Clinical trial data are extremely limited, and their conditions were fundamentally different from animal experiments. There is extensive literature describing mitochondrial transfer/transplantation, introducing novel techniques and methods regarding mitochondrial transplantation [80,99]. However, some works in the literature have been called into question and disputed in some detail, such as the validity of transplantation, the methods of mitochondrial administration, and exogenous mitochondrial compatibility. Despite the excellent efficacy and clinical case studies, it is impossible to explain how a very small number of mitochondria penetrating into a cell can compensate for the dysfunction of numerous endogenous mitochondria. Alternative mechanisms of mitochondrial transplantation remain insufficiently studied. The efficiency and safety of AMT/T in treating cardiovascular diseases needs to be further evaluated.

Thanks to the intense investigation and tangible results in the development of strategies, the successful treatment of patients with these tremendously difficult conditions may not be so far away. The current limited treatment regimens alleviate primary mitochondrial disorders, but there is potential for emerging technologies, in particular, those involving direct manipulation of the mitochondrial genome, to treat this class of diseases more decisively. Nonetheless, the common notion that mitochondrial dysfunction has no cure is currently being challenged by scientists and clinicians.

## 7. Conclusions

Due to MCM’s high level of clinical, genetic, and biochemical diversity, the small number of patients, and a lack of acceptable preclinical models, the definition of beneficial clinical outcomes and the development of effective medications is challenging. Nonetheless, continuous advancement in our knowledge of the molecular basis underlying mitochondrial biogenesis in physiological and pathological conditions is being pursued, which will help elucidate novel mechanistic pathways and discover novel therapies that can prevent the onset and progression of heart failure, thereby advancing a new era of personalized therapeutics and improving health outcomes for patients with MCM.

## Figures and Tables

**Figure 1 cells-11-03511-f001:**
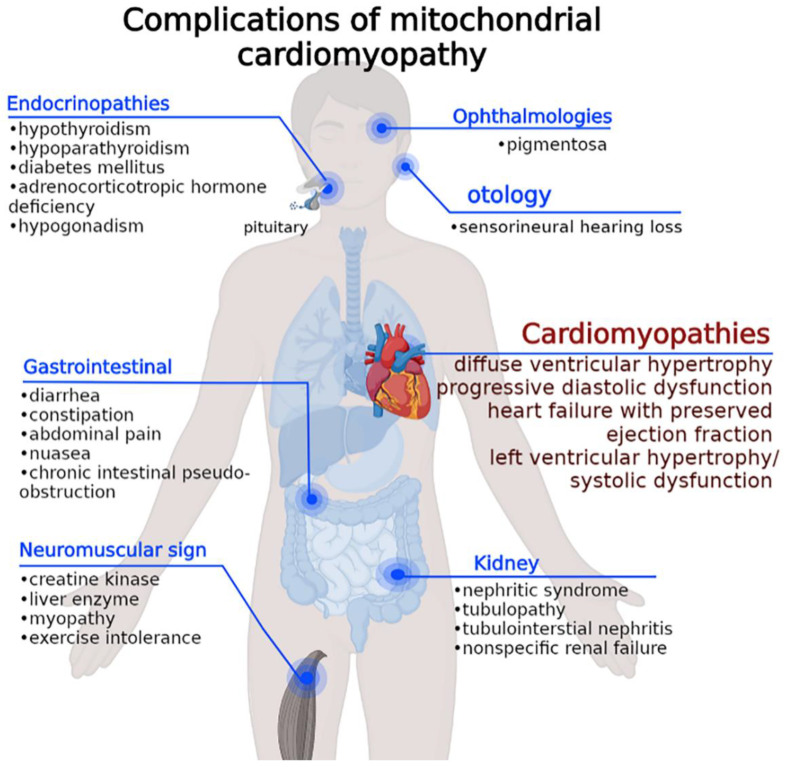
Common complications of MCM. MCM patients can present a range of systemic multiorgan symptoms. The spectrum of tissues involved varies between the mutation (mtDNA or nDNA), heteroplasmy, and age of onset and thus makes it difficult to predict disease progression.

**Figure 2 cells-11-03511-f002:**
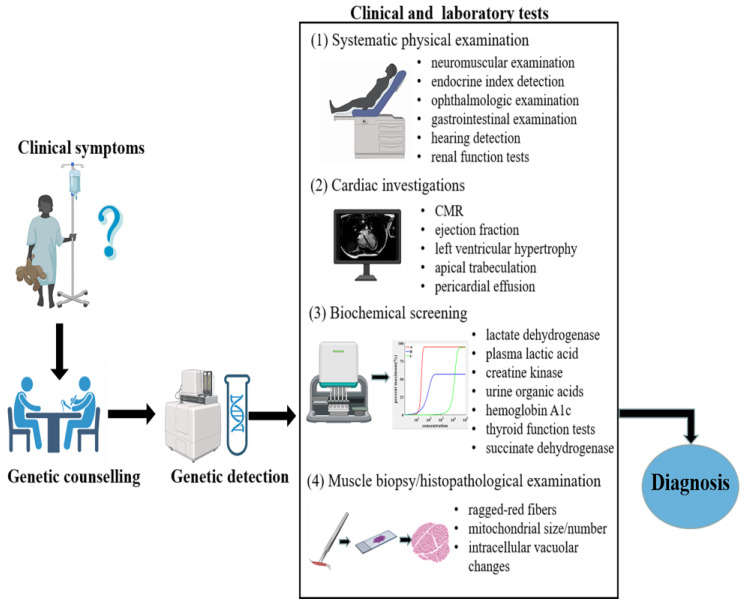
Flow chart for diagnosis of mitochondrial diseases.

**Table 1 cells-11-03511-t001:** List of known causative genes and variants of mitochondrial cardiomyopathy (MCMs).

a. Gene Mutations in Mitochondrial DNA and Mitochondrial Disorder			
Genes	Amino Acid Change	Cardiological Phenotype	Other Disorder
MT-ND1: m. 3481G > A	p. Glu59Lys	HCM, LVNC	LHON
MT-ND4: m. 11778G > A	p. Arg340His	DCM	LHON, progressive dystonia
MTND5: m. 12338T > C	p. Met1Thr	HCM. WPW	Leigh syndrome
MT-ATP6/8: m. 8528 T > C	p. Pro10Ser	HCM	Subacute encephalopathy
MT-ATP6: m. 8851 T > C	p. Trp109Arg	HCM	NARP, Leigh disease
MT-ND6: m. 14453G > A	p. tRNALeu	HCM	LHON, MELAS
MT-ND6: m. 8528 T > C	Syn	DCM	LHON, MELAS
MT-CYB: m. 14849T > C	p. Ser35Pro	HCM	Septo-optic dysplasia
MT-TL1: m. 3260A > G	p. tRNALeu (UUR)	HCM, RCM, LVNC	MELAS, Leigh syndrome
MT-TI: m. 4300A > G	-	HCM, DCM	MERRF, Leigh syndrome
MT-TV: m. 1644G > A	-	HCM	MERRF, Leigh syndrome
MT-TK: m. 8344A > G	p. tRNALys	HCM	MERRF, Leigh syndrome
MT-RNR1: m. 1555A > G	-	RCM	Maternally inherited deafness
**b. Genes Mutations in Nuclear DNA and Mitochondrial Disorder**			
Genes	Amino acid change	Cardiological phenotype	Other disorder
NDUFS2: c. 208 + 5G > A	p. Pro229Gln	HCM	Mitochondrial CI deficiency
NDUFV2: c. 669_670insG	p. Ser224fs	HCM	Mitochondrial CI deficiency
NDUFA11: c. 99 C + 5G> A	p. Ala132Pro	HCM	Mitochondrial CI deficiency and/or encephalocardiomyopathy
NDUFB11: c. 136_142dup	p. Arg134Ser	LVNC, WPW	Mitochondrial CI deficiency
SDHD: c. 275A > G	p. Asp92Gly	DCM, LVNC	Mitochondrial CII deficiency
NDUFAF1: c. 631C > T	p. Arg211Cys	HCM	Mitochondrial CI deficiency
ACAD9: c. 797G > A	p. Arg266Gln	HCM	Mitochondrial CI deficiency
SCO2: c. 418G > A	p. Glu140Lys	HCM	Cytochrome C oxidase deficiency
COX10: c. 610A > G	p. Asn204Asp	HCM	Mitochondrial CIV deficiency
COX15: c. 1129A > T	p. Lys377x	HCM	Cytochrome C oxidase deficiency
COA6: c. 196 T > C	p. Trp66Arg	HCM	Mitochondrial CIV deficiency
COX6B1: c. 58C > T	p. Arg20Cys	HCM	MELAS, MERRF
TEME70: c. 366A > T	p. Tyr112Ter	HCM	Mitochondrial CV deficiency
TEME70: c. 317-2A > G	-	HCM	Mitochondrial CV deficiency
AARS2: c. 1774C > T	p. Arg958 *	HCM	COXPD 8
MRPS22: c. 644T > C	p. Leu215Pro	HCM	COXPD 8
MRPL3: c. 950C > G	p. Pro317Arg	HCM	COXPD9
MRPL3: c. 49delC	Arg17Aspfs * 57	HCM	COXPD9
MRPL44: c. 467T > G	p. Leu156Arg	HCM	Mitochondrial CIV deficiency
TSFM: c. 355G > C	p. Val119Leu	HCM, DCM	COXPD 3
GTPB3: c. 1291dupC;	p. Pro430Argfs * 86	HCM, DCM	COXPD23, Encephalopathy
GTPB3: c. 1375G > A	p. Glu459Lys	HCM, DCM	COXPD23
GTPB3: c. 476A > T	p. Glu159Val	HCM, DCM	lactic acidosis, leukodystrophy
GTPB3: c. 964G > C	p. Ala322Pro	HCM, DCM	lactic acidosis, leukodystrophy
MTO1: c. 1282G > A	p. Ala428Thr	HCM	COXPD10
MTO1: c. 1858dup	p. Arg620Lysfs * 8	HCM	COXPD10
ELAC2: c. 631C > T	p. Arg211 *	HCM	COXPD17
ELAC2: c. 1559C > T	p. Thr520Ile	HCM	COXPD17
ELAC2: c. 460T > C	p. Phe154Leu	MELAS	Cardiac failure
ELAC2: c. 1267C > T	p. Leu423Phe	DCM	Cardiac failure, COX deficiency
TAZ: c. 527A > G	p. His176Arg	DCM, LVNC	BTHS
AGK: c. 306T > G	p. Tyr102Ter	HCM	Sengers syndrome
SLC22A5: c. 12C > G	p. Tyr4 *	HCM, DCM	Systemic primary carnitine deficiency
ACADVL: c. 104delC	p. P35Lfs * 26	HCM, DCM	VLCAD deficiency
ACADVL: c. 848T > C	p. V283A	HCM	VLCAD deficiency
ACADVL: c. 1141_1143del GAG	p. E381del	HCM	VLCAD deficiency
ACAD9: c. 555-2A > G	p. Ala390Thr	HCM	MTP deficiency with myopathy and neuropathy
ATAD3A-C: c. 1064G > A	p. G355D	HCM	Hereditary spastic paraplegia, axonal neuropathy
SLC25A4: c. 239G > A	p. Arg80His	HCM	Mitochondrial DNA depletion syndrome-12
SLC25A4: c. 703C > G	p. Arg235Gly	HCM	Mitochondrial DNA depletion syndrome-12
QRSL1: c. 398G > T	p. G133V	HCM	COXPD40
KARS: c. 1343 T > A:	p. V448D	HCM, DCM, MC	Infantile-onset progressive leukoencephalopathy /or deafness
KARS: c. 953 T > C	p. I318T	HCM, DCM, MC	Mitochondrial cytopathy
TOP3A: c. 298A > G	p. Met100Val	DCM	adult-onset mitochondrial disorder
TOP3: c. 403C > T	p. Arg135Ter	DCM	Adult-onset mitochondrial disorder
FXN: GAA repeat expansion	-	HCM	Friedreich ataxia, MELAS, MERRF
BOLA3: c. 287A > G	p. H96R	HCM	Multiple mitochondrial dysfunctions syndrome-2 with hyperglycinemia
CoQ4: c. 718C > T	p. R240C	HCM	Coenzyme Q10 deficiency 7
CoQ4: c. 421C > T	p. R141X	HCM	Lethal infantile mitochondrial disorder
DNAJC19: IVS3-1G > C	-	DCM, LVNC	3-methylglutaconic aciduria type V

**Abbreviations**: BTHS, Barth syndrome; COXPD, combined oxidative phosphorylation deficiency; COX, cyclooxyganese; DCM, dilated cardiomyopathy; HCM, hypertrophic cardiomyopathy; LHON, Leber’s hereditary optic neuropathy; LVNC, left ventricular non compaction; MC, mitochondrial myopathy; MELAS, mitochondrial myopathy, encephalopathy, lactic acidosis, and stroke-like episodes; MERRF, myoclonus epilepsy associated with ragged red fibers; MTP, mitochondrial trifunctional protein; RCM, restrictive cardiomyopathy; NARP, neurogenic muscle weakness, ataxia, and retinitis pigmentosa; VLCAD, very long-chain acyl-CoA dehydrogenase; WPW, Wolff–Parkinson–White syndrome. “-” mean noncoding. The asterisk denotes the position of the termination codon. The number represents its position in the protein, i.e., 19 means it is in the 19th position from the N terminal to the C terminal.

**Table 2 cells-11-03511-t002:** List of the current animal models used in MCM.

Animal	Mutations	Cardiological Phenotype	Clinical Manifestations	Year	Ref
Mice	Ant1	mitochondrial myopathy/MCM	ragged-red muscle fibers, dramatic proliferation of mitochondria, cardiac hypertrophy with mitochondrial proliferation, severe defects in coupled respiration, metabolic acidosis, exercise intolerance.	1997	[21]
Mice	Ant1	DMC	substantial myocardial hypertrophy/ventricular dilation, cardiac function declining in early age, LV circumferential, radial, rotational mechanics reduced, myocyte hypertrophy, fibrosis, calcification.	2011	[22]
Mice	MRPS34	progressive cardiomyopathy	fractional shortening of the heart, pronounced liver dysfunction, inhibition of mitochondrial translation, decreased oxygen consumption and respiratory complex activity.	2015	[23]
Mice	Med30zg	MCM/cardiac failure	changes in transcription of cardiac genes for OXPHOS and mitochondrial integrity precipitous lethality 2–3 weeks after weaning.	2011	[24]
Mice	Ndufs6	CI deficiency-specific MCM	LV systolic function, cardiac output, and functional work capacity markedly reduced, at increased risk of cardiac failure and death after 4 months in males and 8 in females, ATP synthesis decreased, hydroxyacylcarnitine increased.	2012	[25]
Mice	TFAM	progressive, lethal DCM	elevated ROS production, activated DNA damage response pathway, decreased cardiomyocyte proliferation.	2018	[26]
Mice	TFAM	MCM	critical enzymes in fatty acid oxidation show decreased expression, glycolytic enzymes show increased expression.	2004	[27]
Mice	CHCHD10-^S55L^	MCM	typical ISR^mt^, mitochondrial architecture and function altered in the heart, metabolic pathway changed from oxidative to glycolytic.	2022	[28]
Mice	CHCHD2/CHCHD10	MCM	C2/C10 DKO mice have disrupted mitochondrial cristae, cleavage of the l-OPA1, activation of the ISR^mt^ and development of cardiomyopathy.	2020	[29]
Rat	Isca1	MMDS with cardiomyopathy; MCM	Isca1 HET rats exhibit thin-walled ventricles, larger chambers, cardiac dysfunction and myocardium fibrosis, damaged mitochondrial morphology, enzyme activity and ATP production.	2021	[30]
Dog	QIL1	MCM	cristae abnormalities and cardiac arrhythmias, hyperplastic mitochondrial, cristae rearrangement, electron dense inclusions, lipid bodies in muscle.	2019	[31]
Zebrafish	ndufa7/hhatla	HCM	cardiac functional defects, associated with increased expression of pathological hypertrophy biomarkers ANP and BNP.	2020	[32]
Mongolian gerbils	iron overload	Iron-overload cardiomyopathy	cardiac hypertrophy, increased cardiac output, and normal exercise tolerance at shorter durations, concentric cardiac hypertrophy, cardiac output and exercise capacity were impaired at longer duration.	2002	[33]
Mice	FXN	FRDA- cardiomyopathy	impaired mitochondrial OXPHOS, bioenergetics imbalance, deficit of Fe-S cluster enzymes and mitochondrial iron overload, recapitulated most features of FRDA cardiomyopathy.	2014	[34]
Mice	Bcs1l^p.S78G^	MCM	GRACILE syndrome, growth failure, progressive hepatopathy and kidney tubulopathy,	2019	[35]
Mice	TAZ	BTHS cardiomyopathy	significantly enlarged hearts, ventricular. dilation at 16-weeks of age, lower total CL concentration, abnormal CL fatty acyl composition.	2021	[36]
Mice	C1QBP	MCM	increased oxidative stress, embryonic lethality with the embryonic fibroblast, cardiomyocyte dysfunction.	2017	[37]
Mice	Crif1	MCM	mice suffered from progressive hypertrophy and died from heart failure; mutant mice died within 2 weeks postnatal, showing cardiac hypertrophy associated with mitochondrial dysfunction.	2013	[38]

**Abbreviations:** ANT1, adenine nucleotide translocase 1; ANP, atrial natriuretic peptide; ATP, adenosine triphosphate; BNP, plasma brain natriuretic peptide; BTHS, Bartters syndrome; CHCHD, coiled-helix-coiled-helix; CL, cardiolipin; C1QBP, Complement component 1 Q subcomponent-binding protein; DCM, dilated cardiomyopathy; FRDA, Friedreich’s ataxia; FXN, frataxin gene; HCM, hypertrophic cardiomyopathy; ISR^mt^, mitochondrial integrated stress response; LV, left ventricle; MCM, mitochondrial cardiomyopathy; MMDS, multiple mitochondrial dysfunction syndromes; MRPS34, mitochondrial ribosomal protein of the small subunit 34; OPA1, mitochondrial dynamin like GTPase; OXPHOS, oxidative phosphorylation; ROS, reactive oxygen species; TAZ, tafazzin; TFAM: mitochondrial transcription factor A.

**Table 3 cells-11-03511-t003:** List of the cellular models used in MCM.

Cell Type	Gene	Variants	Disease	Phenotype	Year	Ref.
Immortalized cells	MT-TL1	m. 3243A > G	MELAS syndrome	defective protein synthesis, reduced activities of MRC.	2011	[41]
Immortalized cells	tRNALys	m. 8344A > G mutation	MERRF	increase ROS, oxidative stress, impaired mitochondrial bioenergetics.	2017	[42]
C2C12	TAZ	TAZ-KO	BTHS	mitochondrial deficits, accumulation of MLCL, ROS, production increased, mitochondrial respiration decreased, myocyte differentiation impaired.	2018	[43]
Fibroblasts	tRNA(Leu/tRNA(Lys)	tRNA(Lys)	MELAS/MERRF	mitochondrial membrane potential and respiration rate decreased, incompetent mitochondria assembly, cell volume occupied by secondary lysosomes and residual bodies.	1996	[44]
Fibroblasts	TAZ	Positive-TAZ mutation	BTHS	cardiolipin, phosphatidylcholine, and phosphatidylethanolamine abnormalities in all tissues.	2003	[45]
iPSCs	KCNQ1	R190Q	Long-QT Syndrome	susceptibility to catecholamine-induced tachyarrhythmia, beta-blockade attenuated duration of the action potential prolonged, reduction in I(Ks) current and altered channel activation and deactivation increased.	2010	[46]
iPSCs	mtDNA mutation	m. 3243A > G	MELAS	MELAS-iPSC-derived fibroblasts with high heteroplasmy levels showed defective CI activity, with low heteroplasmy levels showed normal CI activity.	2015	[47]
iPSCs	mtDNA mutation	m. 3243A > G	HMC	neuronal and cardiac maturation defects in iPSC line carrying a quite high proportion of m. 3243A > G, defective mitochondrial respiratory inhibits maturation of iPSC.	2017	[48]
iPSC-CMs	TAZ	c. 328T > C	MCM-BTHS	abnormal metabolic, structural and functional, assembled sparse and irregular sarcomeres.	2014	[49]
iPSC-CMs	HCM-mutation	MYH7/MYBPC3/TNNT2	DD-HCM	impaired diastolic function, prolonged relaxation time, decreased relaxation rate and sarcomere length.	2019	[50]
iPSC-CMs	FXN	rs137854888	FRDA- MCM	disorganized mitochondrial network and (mtDNA) depletion, α-actinin-positive cell sizes and BNP gene expression increased, intracellular iron accumulated, energy synthesis dynamics, ATP production rate impaired.	2014	[51]
iPSC-CMs	FXN	expanded GAA	FRDA	no biochemical phenotype, decreased membrane potential in neurons and progressive mitochondrial degeneration in cardiomyocytes, increased BNP expression and disrupted iron homeostasis.	2013	[52]
iPSC-CMs	MT-RNR2 A	m. 2336T > C	HCM	mitochondrial dysfunctions and ultrastructure defects, ATP/ADP ratio and membrane potential reduced, intracellular Ca^2+^ elevated, electrophysiological abnormalities.	2018	[53]
PBMC-iPSC	C1QBP	c. 823C > T	COXPD	iPSCs express pluripotent markers, have trilineage differentiation potential, carry C1QBP-L275F mutation, and have a normal karyotype.	2014	[54]
iPSC-CMs	DNAJC19	rs137854888	DCMA	highly fragmented and abnormally shaped mitochondria associated with imbalanced isoform ratio of OPA1.	2020	[55]

**Abbreviations**: ADP, adenosine diphosphate; ATP, adenosine triphosphate; BNP, brain natriuretic peptide; BTHS, Barth syndrome; COXPD, combined oxidative phosphorylation deficiency; C1QBP, complement component 1 Q subcomponent-binding protein; DCMA, the dilated cardiomyopathy with ataxia syndrome; DD-HCM, diastolic dysfunction-hypertrophic cardiomyopathy; DNAJC19, mitochondrial import inner membrane translocase subunit TIM14; FXN, frataxin; HCM, hypertrophic cardiomyopathy; MELAS syndrome, mitochondrial encephalomyopathy with lactic acidosis and stroke-like episodes; MLCL, monolyso-cardiolipin; mtDNA, mitochondrial DNA; MT-TL1, mitochondrial mutant gene; OPA1, mitochondrial dynamin like GTPase; TAZ, gene tafazzin; tRNALys, transfer ribonucleic acid lysine.

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
