# Peer review of "Mitochondrial Cardiomyopathy: Molecular Epidemiology, Diagnosis, Models, and Therapeutic Management"

_cells, 2022, doi:10.3390/cells11213511_

Round 1

Reviewer 1 Report

This article reviewed the complex molecular epidemiology of MCM, discussed the current methods of diagnosis, and highlighted the cellular models and animal models that mimics MCM in vivo and in vitro. They emphasize the novel and emerging therapeutics including pharmacological strategies, gene therapies, and mitochondrial replacement therapy or artificial mitochondrial transfer/transplantation. The article addresses an emerging cardiac challenge, information included is important to the filed, and manuscript is well-organized. A few issues should be addressed.

1.       Introduction: Mitochondrial dysfunction and subsequent disruption of ATP production is involved in multiple (if not all) forms of cardiomyopathies and contributes to disease development. The unique pathogenic/pathologic characteristics of mitochondrial dysfunction in MCM comparing with other forms of cardiomyopathies should be better described.  

2.       Molecular epidemiology and multiorgan clinical expression of MCM: This section nicely summarized multiorgan clinical expression of MCM. The “molecular epidemiology” needs to be expanded to include more information of MCM epidemiology at molecular level. Specific genes/proteins that are mutated in MCM patients should be summarized in a table. This is the foundation for “Disease modeling for MCM” listed in Tables 1 and 2.

3.       Current “Disease modeling for MCM” is too long and should be shortened. 

Author Response

Response 1: While many mitochondrial proteins are encoded within the nuclear genome (nDNA), the genetics of mtDNA from which many mitochondrial myopathies arise, produces a heterogeneity of disease, even within the same organ. The possible pathogenic characteristics of mitochondrial dysfunction in MCM is thought to be complex and likely involved multiple abnormal processes in the cells, stemming from deficient oxidative phosphorylation and ATP depletion. The funding by Zhang et al. suggested that mitochondrial function regulates cardiomyocytes proliferation during development and the defective oxidative phosphorylation (OXPHOS), even though did not affect energy supply in embryonic cardiomyocytes, led to excessive ROS generation and inhibition of cell cycle activity(Zhang D, Li Y, Heims-Waldron D, Bezzerides V, Guatimosim S, Guo Y, Gu F, Zhou P, Lin Z, Ma Q, Liu J, Wang DZ, Pu WT. Mitochondrial Cardiomyopathy Caused by Elevated Reactive Oxygen Species and Impaired Cardiomyocyte Proliferation. Circ Res. 2018 Jan 5;122(1):74-87.). Thus, the inhibition of cardiomyocytes proliferation could be a potential mechanism for cardiac dysfunction in mitochondrial disease patients. The possible mechanisms of pathophysiology in mitochondrial dysfucntion-related MCM includes the insufficient energy metabolism in the cardiomyocyte, the abnormal ROS homeostasis, dysfunctional mitochondrial dynamics, abnormal calcium homeostasis, and mitochondrial iron overload. Therefore, individuals with pathogenic variants in the mtDNA of the cardiomyocytes demonstrate myocardial dysfunction(Campbell T, Slone J, Huang T. Mitochondrial Genome Variants as a Cause of Mitochondrial Cardiomyopathy. Cells. 2022 Sep 11;11(18):2835. doi: 10.3390/cells11182835.).

Response 2: Thanks for your advise. I have summarized the specific gene/proteins that are mutated in MCM patients in table 1, as shown below. The original table 1 and table 2 in the manuscirpt were changed into table 2 and table 3, respectively.

Table 1 List of known causative genes and variants of mitochondrial cardiomyopathy (MCMs)

a. Genes Mutations in Mitochondrial DNA and Mitochondrial Disorders

Genes

Amino acid change

Cardiological phenotype

Other disorder

MT-ND1: m.3481G>A

p.Glu59Lys

HCM, LVNC

LHON

MT-ND4: m.11778G>A

p.Arg340His

DCM

LHON, progressive dystonia

MTND5: m.12338T>C

p.Met1Thr

HCM. WPW

Leigh syndrome

MT-ATP6/8: m.8528 T>C

p. Pro10Ser

HCM

Subacute encephalopathy

MT-ATP6: m.8851 T > C

p. Trp109Arg

HCM

NARP, Leigh disease

MT-ND6: m.14453G>A

p. tRNALeu

HCM

LHON, MELAS

MT-ND6: m.8528 T>C

Syn

DCM

LHON, MELAS

MT-CYB: m.14849T>C

p.Ser35Pro

HCM

Septo-optic dysplasia

MT-TL1: m.3260A>G

p. tRNALeu (UUR)

HCM, RCM, LVNC

MELAS, Leigh syndrome

MT-TI: m.4300A>G

-

HCM, DCM

MERRF, Leigh syndrome

MT-TV: m.1644G>A

-

HCM

MERRF, Leigh syndrome

MT-TK: m.8344A>G

p. tRNALys

HCM

MERRF, Leigh syndrome

MT-RNR1: m.1555A>G

-

RCM

Maternally inherited deafness

b, Genes Mutations in Nuclear DNA and Mitochondrial Disorders

Genes

Amino acid change

Cardiological phenotype

Other disorder

NDUFS2: c.208+5G>A

p.Pro229Gln

HCM

Mitochondrial CI deficiency

NDUFV2: c.669_670insG

p.Ser224fs

HCM

Mitochondrial CI deficiency

NDUFA11: c.99 C+5G> A

p. Ala132Pro

HCM

Mitochondrial CI deficiency and/or encephalocardiomyopathy

NDUFB11: c.136_142dup

p. Arg134Ser

LVNC, WPW

Mitochondrial CI deficiency

SDHD: c.275A>G

p.Asp92Gly

DCM, LVNC

Mitochondrial CII deficiency

NDUFAF1: c.631C>T

p.Arg211Cys

HCM

Mitochondrial CI deficiency

ACAD9: c.797G>A

p.Arg266Gln

HCM

Mitochondrial CI deficiency

SCO2: c.418G > A

p.Glu140Lys

HCM

Cytochrome C oxidase deficiency

COX10: c.610A > G

p.Asn204Asp

HCM

Mitochondrial CIV deficiency

COX15: c. 1129A > T

p.Lys377x

HCM

Cytochrome C oxidase deficiency

COA6: c.196 T > C

p.Trp66Arg

HCM

Mitochondrial CIV deficiency

COX6B1: c.58C>T

p.Arg20Cys

HCM

MELAS, MERRF

TEME70: c.366A>T

p.Tyr112Ter

HCM

Mitochondrial CV deficiency

TEME70: c.317-2A>G

-

HCM

Mitochondrial CV deficiency

AARS2: c.1774C > T

p.Arg958*

HCM

COXPD 8

MRPS22: c.644T>C

p.Leu215Pro

HCM

COXPD 8

MRPL3: c.950C>G

p.Pro317Arg

HCM

COXPD9

MRPL3: c.49delC

Arg17Aspfs*57

HCM

COXPD9

MRPL44: c.467T>G

p.Leu156Arg

HCM

Mitochondrial CIV deficiency

TSFM: c.355G>C

p.Val119Leu

HCM, DCM

COXPD 3

GTPB3: c.1291dupC;

p.Pro430Argfs∗86

HCM, DCM

COXPD23, Encephalopathy

GTPB3: c.1375G>A

p. Glu459Lys

HCM, DCM

COXPD23

GTPB3: c.476A>T

p.Glu159Val

HCM, DCM

lactic acidosis, leukodystrophy

GTPB3: c. 964G>C

p. Ala322Pro

HCM, DCM

lactic acidosis, leukodystrophy

MTO1: c.1282G >A

p.Ala428Thr

HCM

COXPD10

MTO1: c.1858dup

p.Arg620Lysfs*8

HCM

COXPD10

ELAC2: c.631C>T

p.Arg211∗

HCM

COXPD17

ELAC2: c.1559C>T

p. Thr520Ile

HCM

COXPD17

ELAC2: c.460T>C

p.Phe154Leu

MELAS

Cardiac failure

ELAC2: c.1267C>T

p.Leu423Phe

DCM

Cardiac failure, COX deficiency

TAZ: c.527A>G

p.His176Arg

DCM, LVNC

BTHS

AGK: c.306T>G

p.Tyr102Ter

HCM

Sengers syndrome

SLC22A5: c.12C>G

p.Tyr4*

HCM, DCM

Systemic primary carnitine deficiency

ACADVL: c.104delC

p.P35Lfs*26

HCM, DCM

VLCAD deficiency

ACADVL: c.848T>C

p.V283A

HCM

VLCAD deficiency

ACADVL: c.1141_1143del GAG

p.E381del

HCM

VLCAD deficiency

ACAD9: c.555-2A>G

p.Ala390Thr

HCM

MTP deficiency with myopathy and neuropathy

ATAD3A-C: c.1064G > A

p.G355D

HCM

Hereditary spastic paraplegia, axonal neuropathy

SLC25A4: c.239G>A

p.Arg80His

HCM

Mitochondrial DNA depletion syndrome-12

SLC25A4: c.703C>G

p.Arg235Gly

HCM

Mitochondrial DNA depletion syndrome-12

QRSL1: c.398G>T

p.G133V

HCM

COXPD40

KARS: c.1343 T>A:

p.V448D

HCM, DCM, MC

Infantile-onset progressive leukoencephalopathy /or deafness

KARS: c. 953 T>C

p.I318T

HCM, DCM, MC

Mitochondrial cytopathy

TOP3A: c.298A>G

p. Met100Val

DCM

adult-onset mitochondrial disorder

TOP3: c. 403C>T

p. Arg135Ter

DCM

Adult-onset mitochondrial disorder

FXN: GAA repeat expansion

-

HCM

Friedreich ataxia, MELAS, MERRF

BOLA3: c.287A>G

p.H96R

HCM

Multiple mitochondrial dysfunctions syndrome-2 with hyperglycinemia

CoQ4: c.718C>T

p.R240C

HCM

Coenzyme Q10 deficiency 7

CoQ4: c. 421C>T

p.R141X

HCM

Lethal infantile mitochondrial disorder

DNAJC19: IVS3-1G>C

-

DCM, LVNC

3-methylglutaconic aciduria type V

Abbreviations: BTHS, Barth syndrome; COXPD, combined oxidative phosphorylation deficiency; COX, cyclooxyganese; DCM, Dilated cardiomyopathy; HCM, Hypertrophic cardiomyopathy; LHON, Leber’s hereditary optic neuropathy; LVNC, Left ventricular non compaction; MC, mitochondrial myopathy; MELAS, mitochondrial myopathy, encephalopathy, lactic acidosis, and stroke-like episodes; MERRF, myoclonus epilepsy associated with ragged red fibers; MTP, Mitochondrial trifunctional protein; RCM, restrictive cardiomyopathy; NARP, neurogenic muscle weakness, ataxia, and retinitis pigmentosa; VLCAD, Very long-chain acyl-CoA dehydrogenase; WPW, Wolff-Parkinson-White syndrome. “-” mean noncoding.

Response 3: Thanks for your advise. I have shortened the “Currect disease modeling for MCM” part by deleting some superfluous expressions, for example,

line 209-210: “. This reduced energetic and functional capacity is consistent with the known susceptibil-ity of individuals with MCM to metabolic crises precipitated by stresses and”;

line 218-222: “It can thus be concluded that at least some of the secondary gene expression alterations in MCM do not compensate but rather directly contribute to heart failure progression [26], [27]. However, the limitation of the model must be taken into account when considering the translational potential of these findings. TFAM deletion is embryonic lethal, and mu-tations of similar severity are unlikely seen in live birth.” Retaining the[26] and [27];

line 232-237: “Researchers identified a variant in QIL1 that resulted in a MCM characterized by cristae abnormalities and cardiac arrhythmias in a canine model. Zebrafish have also proven useful to model human heart diseases due to similarity of their hearts and readily availa-ble genetic methods. Iron-overload cardiomyopathy is the most common cause of death in patients with thalassemia major, yet the associated changes in cardiac function remained unknown.”, transfer [31] and [32] to the dog and zebrafish in line 232;

line 261: “as animal models”;

line 305: “-derived from mouse skeletal myoblast cells”;

line 332-333: ” indicating that iPSCs can be models for mitochondrial diseases”;

line 338: “derived from patients”;

Correction:

In table 2 “List of the current animal model using in MCM”. The No. of insert reference of the last two citations were wrong(40 and 41) and have been modified to 37 and 38.

Reviewer 2 Report

This review summarizes findings of molecular epidemiology of MCM, diagnosis, cellular models, animal models, the pathogenesis of MCM, and therapies. Yang et al. also mentioned current and experimental pharmacological and non-pharmacological therapeutics.

The review is well structured and easy to orient in this field.

I have only a minor suggestion.

I recommend listing all abbreviations mentioned in Tables (1 and 2) in alphabetical order. Also, include all missing abbreviations.

In summary, I consider this paper suitable for published after a minor revision.

Author Response

Response 1: Thank you for your advise. I have listed all abbreviations in alphabetical order in tables. 

Table 2. List of the current animal model using in MCM.

Abbreviations: ANT1, adenine nucleotide translocase 1; ANP, atrial natriuretic peptide; ATP, adenosine triphosphate; BNP, plasma brain natriuretic peptide; BTHS, Bartters syndrome; CHCHD, coiled-helix-coiled-helix; CL, cardiolipin; C1QBP, Complement component 1 Q sub-component-binding protein; DCM, dilated cardiomyopathy; FRDA, Friedreich’s ataxia; FXN, frataxin gene; HCM, hypertrophic cardiomyopathy; ISRmt, mitochondrial integrated stress re-sponse; LV, left ventricle; MCM, mitochondrial cardiomyopathy; MMDS, multiple mitochon-drial dysfunction syndromes; MRPS34, mitochondrial ribosomal protein of the small subunit 34; OPA1, mtochondrial dynamin like GTPase; OXPHOS, oxidative phosphorylation; ROS, reactive oxygen species; TAZ, tafazzin; TFAM: mitochondrial transcription factor A; VMHC, ventricular myosin heavy chain.

Table 3. List of the cellular models using in MCM.

Abbreviations: ADP, adenosine diphosphate; ATP, adenosine triphosphate; BNP, brain natriu-retic peptide; BTHS, Barth syndrome; COXPD, combined oxidative phosphorylation deficiency; C1QBP, complement component 1 Q subcomponent-binding protein; DCMA, the dilated car-diomyopathy with ataxia syndrome; DD-HCM, diastolic dysfunction-hypertrophic cardiomyo-pathy; DNAJC19, mitochondrial import inner membrane translocase subunit TIM14; FXN, frataxin; HCM, hypertrophic cardiomyopathy; MELAS syndrome, mitochondrial encephalo-myopathy with lactic acidosis and stroke-like episodes; MLCL, monolyso- cardiolipin; mtDNA, mitochondrial DNA; MT-TL1, mitochondrial mutant gene; OPA1, mitochondrial dynamin like GTPase; TAZ, gene tafazzin; tRNALys, transfer ribonucleic acid lysine.

Reviewer 3 Report

The review paper titled “Mitochondrial Cardiomyopathy: Molecular Epidemiology, Diagnosis, Models, and Therapeutic Management” is very interesting. The authors discussed how changes in mitochondria (mutations) will lead to mitochondrial cardiomyopathy. They highlighted the in vivo and in vitro models used for modeling cardiomyopathy.

Major comments –

1)      Can authors Provide flow chart for diagnosis of mitochondrial disease. (Approach to make the diagnosis).

2)       Why did not authors discuss different techniques involved in Mitochondrial replacement therapy? (Like spindles transfer, pronuclear transfer, polar body transfer).

3)      There are numerous Spelling mistakes, for example in figure – under cardiomyopathies –“Ventricular” is written as Wentricular.

Author Response

Response 1: Thank you for your advise. I have drawn a flow chart for diagnosis of mitochondrial disease shown as below (Figure 2).

Figure 2. The flow chart for diagnosis of mitochondrial disease

Response 2: Mitochondrial replacement therapy (MRT) was mentioned as a new, promising, and controversial technique in the treatment of mitochondrial cardiomyopathy. Spindles transfer, pronuclear transfer, and polar body transfer are three main techniques involved in MRT,  considering the the length of the article, the specific techniques was not discussed in detail in this review article.

Response 3: Thans a lot. I have checked the spelling mistakes of the figure 1 and the whole text and corrected them.

Figure 1. The common complications of MCM

Round 2

Reviewer 1 Report

The authors addressed previous concerns. 

Reviewer 3 Report

The author has addressed all the comments and provided substantial information. It is very good article and definitely worth of publishing.